# Characterization of Linezolid-Analogue L3-Resistance Mutation in *Staphylococcus aureus*

**DOI:** 10.3390/microorganisms11030700

**Published:** 2023-03-08

**Authors:** Anna Zanfardino, Michela Di Napoli, Federica Migliore, Bruno Hay Mele, Annunziata Soriente, Margherita De Rosa, Eugenio Notomista, Mario Varcamonti

**Affiliations:** 1Department of Biology, University of Naples Federico II, Via Cintia, 80126 Naples, Italy; 2Department of Chemistry and Biology “Zambelli”, University of Salerno, Via Giovanni Paolo II, 132, 84084 Fisciano, Italy

**Keywords:** linezolid analogue, antibiotic resistance, antimicrobial activity

## Abstract

In a previous study, a linezolid analogue, called 10f, was synthesized. The 10f molecule has an antimicrobial activity comparable to that of the parental compound. In this study, we isolated a *Staphylococcus aureus* (*S. aureus*) strain resistant to 10f. After sequencing the 23S rRNA and the ribosomal proteins L3 (rplC) and L4 (rplD) genes, we found that the resistant phenotype was associated with a single mutation G359U in rplC bearing to the missense mutation G120V in the L3 protein. The identified mutation is far from the peptidyl transferase center, the oxazolidinone antibiotics binding site, thus suggesting that we identified a new and interesting example of a long-range effect in the ribosome structure.

## 1. Introduction

Multidrug resistance in Gram-positive pathogen bacteria is one of the most significant challenges for the scientific community involved in the research and discovery of new and more effective antimicrobial agents active against these pathogens. Linezolid, an oxazolidinone antibiotic, is effective for the treatment of infections caused by Gram-positive pathogens resistant to other antibiotics including methicillin-resistant *S. aureus* (MRSA), vancomycin-resistant enterococci (VRE), and penicillin-resistant *Streptococcus pneumoniae* [1]. Favorable pharmacokinetic and toxic effect profiles, consistent with oral or intravenous administration in humans, represent significant features which make linezolid an antibiotic of great success [2], also showing several characteristics appropriate to reduce the occurrence of drug resistance.

Indeed, linezolid is a completely synthetic drug; thus, no natural and pre-existing pool of resistance genes would be expected to ease the appearance of resistance mechanisms. Furthermore, it has a unique mechanism of action which targets bacterial protein synthesis at an extremely early stage [3], and, consequently, cross-resistance between the drug and commercially available antimicrobials would be remote.

In any case, the identification of linezolid-resistant bacteria [4] has already underlined the need to find new oxazolidinone-type drugs with different targets that bypass resistance. Studies of new oxazolidinone with structural changes and improved features are underway and the research area is very active [5]. In a previous paper [6], we have described the design, the synthesis and the preliminary anti-bacterial activity of unreported linezolid analogues bearing the urea and thiourea functionality at the C-5 position. In this paper, we describe the anti-microbial activity of one of these linezolid analogues, called 10f. To understand the mechanism of action of this analogue, resistant mutants of *S. aureus* were generated.

## 2. Materials and Methods

### 2.1. Minimum Inhibitory Concentration (MIC) Determination

Minimal inhibitory concentrations (MICs) for linezolid and its various analogues were determined according to the microdilution method established by the Clinical and Laboratory Standards Institute. 10f was tested at different concentrations ranging from 36 to 0.06 mg/mL. Five microliters of a bacterial suspension were added to each well of a 96-well microtiter plate to yield a final concentration of about 5 × 10^5^ cells/mL. MICs were determined after 16 and 20 h of incubation at 35 ± 2 °C [7].

### 2.2. Cytotoxicity Assay

In flat-bottomed 6-well plates, HeLa (cervix adenocarcinoma) and H1299 (human non-small cell lung carcinoma) cells were cultured at a density of 2.5 × 10^5^ cells per well and were supplemented or not with 10f (0.5, 1, 2, 4, 8, 12, and 24 µM) for 8, 24, and 48 h. CellTiter 96^®^ Aqueous One Solution Reagent (Promega, Madison, WI, USA) was added to each well after a 10f treatment, according to the manufacturer’s instructions [8]. Absorbance at 490 nm was determined after 30 min using a Multiskan spectrum (Thermo Electron Corporation. Waltham, MA, USA).

### 2.3. Isolation of Resistant and Revertant Mutant 10f

Five microliters of an overnight bacterial suspension of *S. aureus* ATCC 6538P were used for MIC determination and were inoculated into multiwell plates containing different concentrations (from 1 to 32 μg/mL of 10f). We found that bacterial cells were able to grow at a concentration of 10f of 16 μg/mL. This resistance was present and stable when the strain was taken under antibiotic selection.

To select the revertant phenotype, a resistant strain was grown in absence of 10f. Four independent resistant colonies were propagated in 96-well microtiter plates in an antibiotic-free medium for 50 days. The plates also contained control wells that were not inoculated by cells. A volume of 1.2 µL of each stationary phase culture was transferred daily to 100 µL of fresh medium using a manual-held 96-pin replicator.

### 2.4. Bacterial DNA Extraction

Genomic DNA was extracted from single *S. aureus* isolate colonies that were inoculated in 5 mL of Luria Bertani broth and incubated overnight at 37 °C using a guanidium-thiocyanate-based method [9].

### 2.5. Polymerase Chain Reaction (PCR) Amplification of Individual 23S rRNA, rplD and rplC Genes

Primer couples were designed based on the published *S. aureus* genome N315 (GenBank accession n. NC_002745). For *S. aureus* isolates with 6 copies of 23S rRNA operons, we used the primers listed in Table 1 (rrn1–rrn6) [10]. PCR conditions were 1 min at 94 °C and 30 cycles of denaturing, annealing, and extension at 94 °C (30 s), 55 °C (30 s), and 72 °C (5 min). PCR products were separated by agarose (1.5%) gel electrophoresis. The 6 individual bands were then gel-extracted and purified (Qiagen). For each purified rRNA gene fragment, the domain V region spanning 2280–2699 bp (*Escherichia coli* numbering) was amplified. The primers used were 5_-GCGGTCGCCTCCTAAAAG-3_ (upper primer, corresponding to bases 2280–2297 of *S. aureus* 23S rRNA gene; GenBank accession no. X68425) and 5_-ATCCCGGTCCTCTCGTACTA-3_ (lower primer, complementary strand corresponding to bases 2680–2699 of *S. aureus* 23S rRNA gene, GenBank accession no. X68425). PCR conditions were 5 min of lysis and denaturation at 94 °C; 30 cycles of denaturing, annealing, and extension at 94 °C (30 s), 55 °C (30 s), and 72 °C (1 min), respectively; and a final 10 min extension at 72 °C. The products were ∼390 bp in size and were separated by agarose (1.5%) gel electrophoresis. PCR conditions for rplD gene amplification were 5 min of lysis and denaturation at 94 °C; 30 cycles of denaturing, annealing, and extension at 94 °C (30 s), 44 °C (30 s), and 72 °C (1 min) respectively; and a final 10 min extension at 72 °C. The products were ∼624 bp in size and were separated by agarose (1.5%) gel electrophoresis. PCR conditions for rplC gene amplification were 5 min of lysis and denaturation at 94 °C; 30 cycles of denaturing, annealing, and extension at 94 °C (30 s), 50 °C (30 s), and 72 °C (1 min), respectively; and a final 10 min extension at 72 °C. The products were ∼663 bp in size and were separated by agarose (1.5%) gel electrophoresis. The PCR products were gel-extracted and purified (Qiagen). They were then sequenced by use of the standard dideoxynucleotide method (Molecular Biology Core Facility, Dana-Farber Cancer Institute; Boston, MA). Sequence data were analyzed by the use of MEGALIGN (DNASTAR) and CHROMAS (version 1.45; Conor McCarthy, School of Health Sciences, Griffith University, Gold Coast Campus; Southport, Queensland, Australia).

### 2.6. Modelling

Docking of 10f in the structure of the large ribosomal subunit from *S. aureus* bound to linezolid (PDB code 4WFA) was performed as previously described [6]. For comparison, numbering of the rRNA form *H. marismortui* has been used through the text.

## 3. Results

### 3.1. 10f Antimicrobial Activity

In order to find antibiotics active against linezolid resistant strains, a new series of 5-substituted oxazolidinones derived from linezolid, having urea and thiourea moieties at the C-5 side chain of the oxazolidinone ring, were tested in a previous paper [6]. The 10f compound (Figure 1B) demonstrated antimicrobial activity comparable to that of linezolid (Figure 1A) against *S. aureus*, recording an MIC value of 1 µg/mL.

In the present study, we have tested 10f against different Gram-positive strains. Table 2 lists the minimum inhibitory concentration (MIC) for linezolid and 10f against different ATCC strains of *S. aureus*, *Enterococcus faecalis*, *Enterococcus facium*, *S. epidermidis* and one *S. aureus* methicillin-resistant strain (WKZ-2). These results showed that the activity of 10f was strongly comparable to that of linezolid against tested strains.

Compound 10f, like linezolid, did not induce significant changes in cell viability for two different eukaryotic cell lines (Hela and H1299), in the concentration range tested and for the exposure times described in the materials and methods section.

### 3.2. Isolation of S. aureus Mutant Resistant to 10f

To characterize the mechanism of action of 10f, we tried to isolate *S. aureus* mutants showing resistance to this compound. *S. aureus* ATCC 6538P 10f-susceptible strain (MIC 1 µg/mL) was grown at 37 °C in Mueller–Hinton broth and then serially passaged in a medium containing increasing concentrations (1 to 32 mg/L) of 10f. During these passages, one *S. aureus* descendant was isolated, which showed a 10f MIC increase to 16 µg/mL. The 23S rRNA genes of this resistant mutant were amplified and sequenced as previously described [10,11]. None of the six 23S genes were mutated. As a number of 50S large-subunit ribosomal proteins have regions which interact closely with the oxazolidinone binding site in the peptidyl transferase center (PTC), to identify the mutated locus responsible for 10f resistance we sequenced two genes coding for ribosomal proteins that were already described to induce linezolid resistance: L4 and L3. L4 belongs to a conserved family of r-proteins with mixed α-helices and β-strands [12]; it is essential for the early steps of ribosome assembly of both bacteria and eukaryotic cells [13,14]. In mature ribosome structures, its globular body domain overlaps the external moieties of domains I and II, while its internal loop region fits deep into the same domains, also reaching parts of the peptidyl transferase center (PTC) in domain V [15]. The resistant strain that we isolated did not show mutations in the gene coding for protein L4 [16]. Therefore, we decided to focus on the gene encoding for L3 ribosomal protein. Mutations in L3 have been associated with resistance against tiamulin (TIA) and retapamulin (whose binding site overlaps with that of oxazolidinones in the PTC) [17,18,19]. However, different researchers described a variety of L3 mutations in *S. aureus* following in vitro selection with oxazolidinones [20]. In our case, we found the mutation G359U in the rp C gene corresponding to a missense mutation at position 120 in the L3 protein that causes a change from Glycine (wild type) to Valine (resistant). Very interestingly, in three different experiments, we propagated (see methods) the *S. aureus*-resistant populations in the absence of 10f for 50 transfers (approximately 400 generations) by diluting 1% of the saturated cultures into fresh medium every 24 h, and we selected a representative clone for further analysis. The 10f resistance disappeared together with the mutation in the L3 gene (see Table 3 for MIC values). This finding is a strong indication that the mutation G120V could be the major factor responsible for the resistant phenotype.

### 3.3. Interaction Model between 10f and the Bacterial Ribosome

We have previously reported a docking analysis of 10f in the structure of the *Haloarcula marismortui* ribosome (PDB code 3CPW) [6]. We suggested that the peculiar syn-anti conformation of the nitrophenyl-thiourea moiety in 10f fits very well with the linezolid pocket, allowing several additional van der Waals and polar interactions, which could counterbalance the loosening effects of mutations conferring resistance to linezolid (e.g., G2447U, U2500A and G2576U). Recently, Eyal and co-workers published the crystal structure of the large ribosomal subunit from *S. aureus* bound to linezolid [21]; therefore, we repeated the docking analysis using this structure as reference (PDB code 4WFA). The structures of the complexes between linezolid and the large ribosomal subunits from *H. marismortui* and *S. aureus* are very similar but not identical. The main difference is in the orientation of the acetamide moiety. In the ribosome from *S. aureus*, it points toward a small pocket defined by G2447 and C2501. In the ribosome from *H. marismortui*, it is folded onto the oxazolidinone ring pointing toward the formyl-Phe-CCA ligand; an analogue of the formyl-Met-tRNA and the small pocket in this structure hosts a potassium ion. Whether the difference is caused by the presence of the formyl-Phe-CCA ligand or not, in the case of 10f, the nitrophenyl-thiourea moiety is too large to adopt the same orientation of the acetamide group observed in the structure of the *S. aureus* ribosome (Figure 2). Indeed, the docking of 10f in the ribosome of *S. aureus* suggests that it should adopt an orientation very similar to that found in the case of the ribosome of *H. marismortui* (Figure 2, Figure 3A and Figure 4A).

In particular, the nitrophenyl–urea moiety makes van der Waals contacts with A2503, G2505, and A2059. Moreover, the nitro group is involved in a H-bond with the N6 of A2058 (Figure 2B and Figure 4A). The additional and extended van der Waals contacts of the nitrophenyl-urea moiety with G2505 are particularly interesting as the base of this nucleotide is staked on the base of G2576 that is mutated to U in several linezolid resistant strains [22] (Figure 2, Figure 3 and Figure 4). Likely, the substitution of a purine with a pyrimidine at position 2576 makes the surroundings of G2505 more flexible, thus reducing the interaction with linezolid. In the case of 10f, the additional stacking interaction between the nitrophenyl ring and G2505 (Figure 2B, Figure 3A and Figure 4A) would counterbalance the increased mobility of this nucleotide, thus allowing a strong interaction of 10f with the mutated ribosome.

The model of the complex 10f/ribosome also provides a possible explanation to the effects of the mutation G120V in the L3 protein. As shown in Figure 3 and Figure 4, L3 has an unusually long loop (spanning from T119 to V178) with an extended beta-structured stem that penetrates into the ribosome and approaches the region, including G2576. In particular, C2575, G2577, C2578, and U2579 are at direct van der Waals contact with several residues in the middle part of the loop, namely G144, S145, H146, F147, G152, S153, G155, M156, and A157 (Figure 4B).

Mutation G120V is at the base of the loop (Figure 3 and Figure 5) and, due to the considerably higher volume of the valine side chain (Figure 5), it would likely cause a rearrangement of the structure that would propagate to the tip of the loop, thus indirectly influencing the mobility of the C2575-U2780 region and hence of the linezolid/10f binding pocket. In particular, one could speculate that the mutation G120V could indirectly cause a repositioning of G2576 and of the adjacent G2505 which in turn would occupy, at least in part, the cavity where the nitrophenyl moiety of 10f is hosted, thus decreasing its binding affinity. Therefore, G120V would be a very interesting example of long-range effects in the complex ribosome structure. It is worth noting that the suggested changes would influence only or mainly the nitrophenyl-thiourea binding pocket, thus selectively impairing the binding of 10f with respect to linezolid and explaining why the mutation G120V does not affect linezolid’s activity.

## 4. Discussion

The spreading of strains resistant to linezolid proves that bacteria can develop resistance in a few years, even against completely artificial antimicrobials. This makes it essential not only to continue searching for new antibiotics but also to study the molecular strategies that bacterial cells adopt to develop resistance in order to design more lasting antibiotics.

In our case, by growing *S. aureus* in the presence of increasing concentrations of the linezolid derivative 10f, we were able to isolate a resistant strain which revealed a very interesting mutation in the ribosomal protein L3. The observed mutation, G120V, to the best of our knowledge, has never been reported before, neither in linezolid or other antibiotics resistant *S. aureus,* nor in other resistant bacteria. As the site of the mutation is quite far from the linezolid and 10f binding site, i.e., the peptidyl transferase center, the observed mutation likely has long-range effects on the shape and/or dynamics of this essential site of the ribosome. As described in detail above, a close inspection of the structure of the ribosome suggests that the mutation G120V might cause a small rearrangement of a long loop which, protruding from the body of L3, penetrates into the ribosome, contacting some of the nucleotides that line up the peptidyl transferase center. This could cause a reduction in the cavity necessary to host the bulky nitrophenyl–thiourea moiety which characterizes 10f. It is worth noting that most of the known mutations in L3 associated with linezolid resistance are located in the central part of the mentioned loop, hence considerably nearer to the peptidyl transferase center than the residue at position 120 [23,24]. Even more interestingly, the mutation G120V in the L3 gene reverted in three different replicated experiments when the mutated strain was grown in the absence of 10f and the reversion of the mutation was accompanied by the disappearance of the resistant phenotype. These findings demonstrate that the observed mutation, even if advantageous in the presence of 10f, decreases the biological fitness of *S. aureus,* thus making its spreading among the population unlikely. Very interestingly, our findings suggest that targeting the cavity which hosts the nitrophenyl–thiourea moiety of 10f is a promising strategy to develop further linezolid derivatives with a lower potential to induce the insurgence of resistant strains.

## Figures and Tables

**Figure 1 microorganisms-11-00700-f001:**
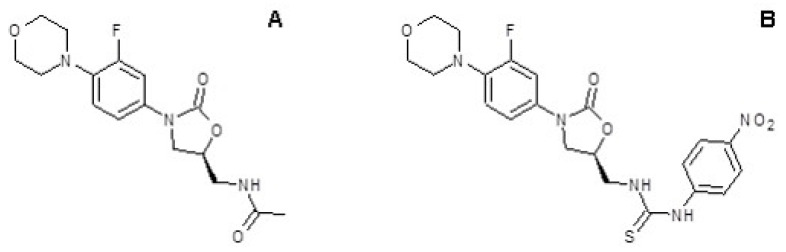
Chemical structures of linezolid and its analogue. Chemical structure of linezolid (**A**) and compound 10f (**B**).

**Figure 2 microorganisms-11-00700-f002:**
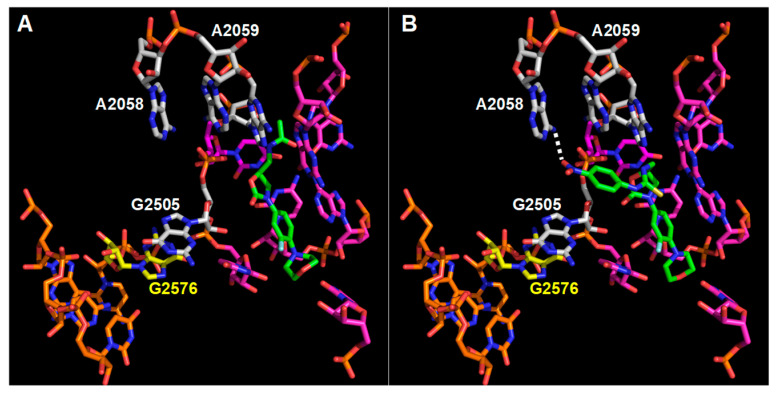
Comparison between the orientations of linezolid and 10f in the large subunit of *S. aureus* ribosome. (**A**) Linezolid binding pocket in the crystal structure of the ribosome/linezolid complex (PDB code 4WFA). (**B**) Model of the complex 10f/large ribosomal subunit. Linezolid, 10f, and the surrounding nucleotides are shown as sticks. Nitrogen atoms are shown in blue, oxygen in red, fluorine in pale cyan, carbon atoms of linezolid and 10f are in green, carbon atoms of the nucleotides of the linezolid binding pocket are in magenta, carbon atoms of the nucleotides which contact only 10f or make more extended contacts with 10f than with linezolid are in white, carbon atoms of G2576 are in yellow and carbon atoms of its neighbors (C2575, G2578, C2579, and U2580) are in orange. The dashed white line in panel (**B**) indicates the possible H-bond between the nitro group of 10f and the NH_2_ group at position 6 of A2058.

**Figure 3 microorganisms-11-00700-f003:**
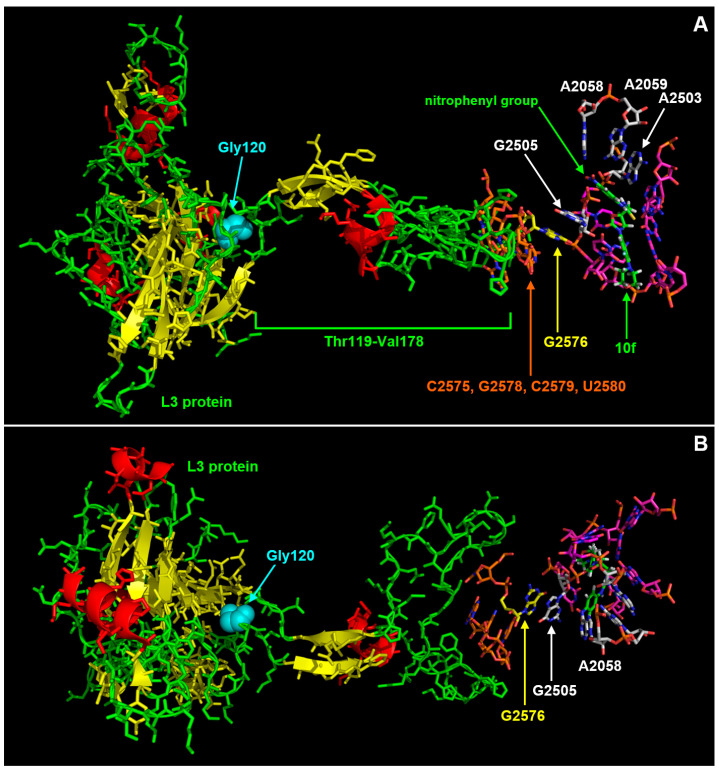
Molecular contacts between L3 and the nucleotides in the linezolid binding pocket. The image in panel (**B**) is rotated by 90° along the left–right axis with respect to (**A)**. L3 is shown as sticks and cartoon images except for residue Gly120, shown as spheres. 10f and the surrounding nucleotides are shown as sticks. L3 is colored according to the secondary structure (helices, red; strands, yellow; loops, green) except Gly120, shown in cyan. In the case of 10f and of the surrounding nucleotides the color code is the same as in Figure 2.

**Figure 4 microorganisms-11-00700-f004:**
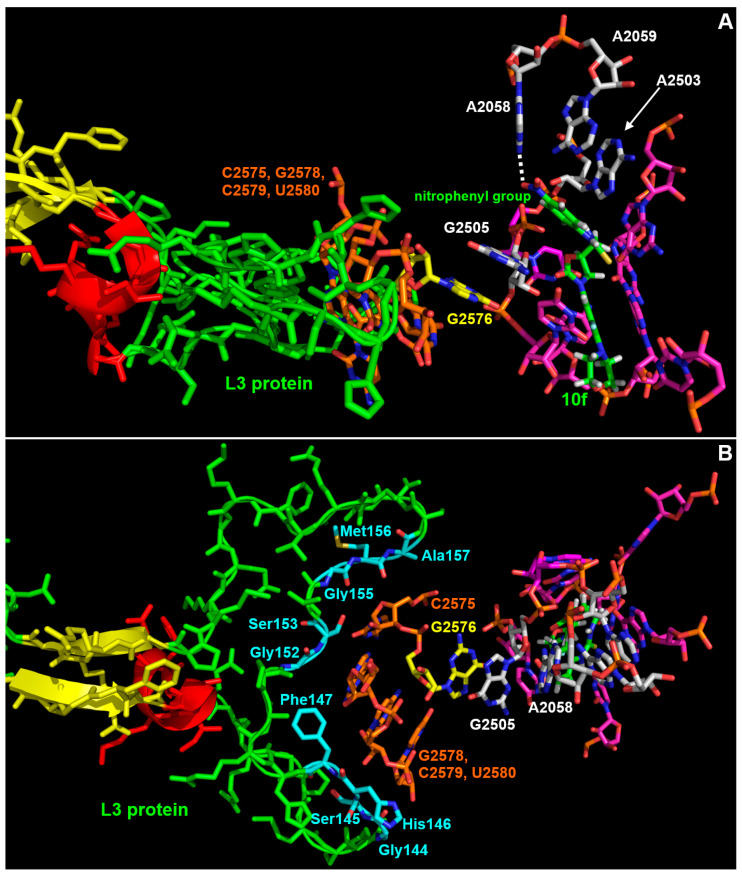
Close-up of the linezolid binding pocket in the model of the complex 10f/large ribosomal subunit. The color scheme is the same as in Figure 3 except that, in panel (**B**), the residues contacting nucleotides C2575, G2578, C2579 or U2580 are colored by atom type (nitrogen atoms are shown in blue, oxygen in red, carbon atoms in cyan). The dashed white line in panel (**A**) indicates the possible H-bond between the nitro group of 10f and the NH_2_ group at position 6 of A2058.

**Figure 5 microorganisms-11-00700-f005:**
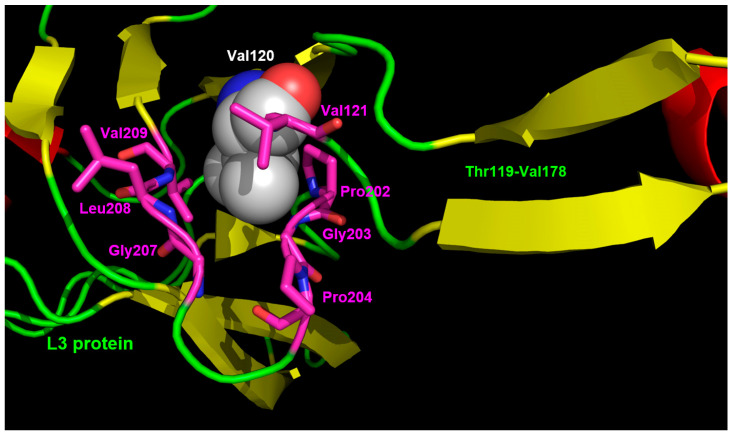
Close-up of the L3 region surrounding position 120. A valine residue is shown at position 120 to highlight the close contacts with residues 202–204 and 207–209. Residues are colored by atom type: nitrogen atoms are shown in blue, oxygen in red, carbon atoms of Val120 are in light gray, and carbon atoms of residue at close contact with Val120 side chain are in magenta.

**Table 1 microorganisms-11-00700-t001:** Oligonucleotides used in PCR reactions.

Primer	Sequence
For-rrn-all	5′-CCCGCACGAAAGGCGTAACG-3′
Rev-rrn1	5′-CGCTTCTTCTTGATTTAAACTTTC-3′
Rev-rrn2	5′-GGCTTTGGCGGTATAGACG-3′
Rev-rrn3	5′-GGCAGATGCTCTCCCAGC-3′
Rev-rrn4	5′-CAACGTGTAAATCATTTCGATC-3′
Rev-rrn5	5′-GGCTCCACAGGTAGGACTCG-3′
Rev-rrn6	5′-CCTGAGCCAGGATCAAAC-3′
Rev-rrn-all	5′-GTTGGGAAATCTCATCTTG-3′
rplC-For	5′-ATGACCAAAGGAATCTTAGG-3′
rplC-Rev	5′-TTATTTATTACCTTTTTTAATTGAAG-3′
rplD-For	5′-ATGGCTAATTATGATGTT-3′
rplD-Rev	5′-TTATCCGAGCACCTCCTC-3′

**Table 2 microorganisms-11-00700-t002:** 10f and linezolid MIC values calculated against different strains.

Strain	MIC 10f (µg/mL)	MIC Linezolid (µg/mL)
*S. aureus* ATCC 29213	2	1
*S. aureus* ATCC 6538P	1	1
MRSA WKZ-2	2	1
*E. faecalis* ATCC 29212	2	1
*E. faecium* ATCC 14434	1	2
*S. epidermidis* ATCC 12228	0.5	0.5

**Table 3 microorganisms-11-00700-t003:** 10f and linezolid MIC values calculated against *S. aureus* wild type (ATCC6538P), 10f resistant (10f^R^) and 10f revertant (10f^rev^) strains.

Strain	MIC 10f (µg/mL)	MIC Linezolid (µg/mL)
*S. aureus* ATCC 6538P	1	1
*S. aureus* 10f^R^	16	1
*S. aureus* 10f^rev^	1	1

## Data Availability

The data presented in this study are available in this article.

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
