# Peer review of "Characterization of Linezolid-Analogue L3-Resistance Mutation in Staphylococcus aureus"

_microorganisms, 2023, doi:10.3390/microorganisms11030700_

Round 1

Reviewer 1 Report

1.     In line 60, “16 e 20h”. what does ‘e’ mean?

2.     In lines 139 to 141, the indication of 10f and linezolid is wrong, according to figure 1.

3.     In line 141, “ug/mL” should be “μg/mL”. And sometimes “μg/mL”, and sometimes “µg/ml” (in table 3). Check through the manuscript to make sure no such mistakes.

4.     In lines 188-189, “we selected a single, representative clone for further analysis the resistance disappeared together with the mutation in the L3 gene”. Check the sentence grammar.

5.     Use stick instead of sphere in all the figures. Spheres make the molecule structure difficult to be recognized.

6.     Add a figure of linezolid’s binding model to show the difference or comparison with 10f.

7.     In the paragraph “Mutation G120V is at the base of the …”, lines 253-261. Add some explanation or suppose to address why G120V has no influence on linezolid’s activity.

8.     In figure 4, a bunch of sticks looks mess. Suggest hiding the irrelevant residues and just showing cartoon.  

Author Response

Thank you for your comments. You'll find the point by point answers in the attached file. All the modified sentences are highlighted in green in the manuscript

In line 60, “16 e 20h”. what does ‘e’ mean?

Thanks for your comment. Text has been changed accordingly.

In lines 139 to 141, the indication of 10f and linezolid is wrong, according to figure 1.

Thanks for your comment. Text has been changed accordingly.

In line 141, “ug/mL” should be “μg/mL”. And sometimes “μg/mL”, and sometimes “µg/ml” (in table 3). Check through the manuscript to make sure no such mistakes

Thanks for the comment. Text has been changed accordingly.

In lines 188-189, “we selected a single, representative clone for further analysis the resistance disappeared together with the mutation in the L3 gene”. Check the sentence grammar.

Thanks for the comment. Text has been rephrased accordingly.

Use stick instead of sphere in all the figures. Spheres make the molecule structure difficult to be recognized.

All figures were modified accordingly with reviewer indication.

Add a figure of linezolid’s binding model to show the difference or comparison with 10f.

A new figure (numbered as “2”) has been added in the manuscript.

In the paragraph “Mutation G120V is at the base of the …”, lines 253-261. Add some explanation or supposed to address why G120V has no influence on linezolid’s activity. An explanation to the reviewer request has been added in the text at the end of 3.3 paragraph (indicate with red characters)

In figure 4, a bunch of sticks looks mess. Suggest hiding the irrelevant residues and just showing cartoon. 

Changes were made accordingly with reviewer request. The modified figure is now number “5”

Reviewer 2 Report

I have reviewed an article: Characterization of linezolid-analogue L3-resistance mutation 2 in Staphylococcus aureus

Subjected topic is worth of investigation and I think this article can be accepted for publication after few minor corrections:

What was rationale to test  on s HeLa and H1299 cells?

What is stability of 10f during 48 hrs of incubation?

Results:

 Table 2: please explain what MIC 0.5-2 means.

Table 3. Please explain the meaning of the values written in the table.

Author Response

Thank you for your comments. You'll find the point by point answers in the attached file. All the modified sentences are highlighted in green in the manuscript

What was rationale to test  on s HeLa and H1299 cells? 

Thanks for the comment. These cell lines are routinely used for cytotoxicity tests and they were available in our lab.

What is stability of 10f during 48 hrs of incubation?

After 72 hrs at room temperature the antibiotic effect of 10f is still present with same efficiency

     Table 2: please explain what MIC 0.5-2 means.

Thanks for the comment. The MIC reported in Table 2 range from a value of 0.5 μg/mL up to a value of 2 μg/mL.

Table 3. Please explain the meaning of the values written in the table.

Thanks for the comment. The MICreported in Table 3 range from a value of 1 μg/mL up to a value of 16 μg/mL.
